# Retrospective analysis of spatiotemporal variation of scrub typhus in Yunnan Province, 2006–2022

Zhuo Li[1☯], Shuzhen Deng[2☯], Tian Ma[1], Jiaxin Hao[1], Hao Wang[1], Xin Han[1], Menghan Lu[1], Shanjun Huang[1], Dongsheng Huang[3], Shuyuan Yang[4], Qing Zhen[1]*, Tiejun Shui[2]*

**1** State Key Laboratory for Diagnosis and Treatment of Severe Zoonotic Infectious Diseases, Key Laboratory for Zoonoses Research of the Ministry of Education, Department of Epidemiology and Biostatistics, School of Public Health, Jilin University, Changchun, Jilin, PR China, **2** Yunnan Center for Disease Control and Prevention, Kunming, Yunnan, China, **3** Baoshan Prefecture Center for Disease Control and Prevention, Baoshan, Yunnan, China, **4** Kunming center for Disease Control and Prevention, Kunming, Yunnan, China

☯ These authors contributed equally to this work.
* zhenqing@jlu.edu.cn (QZ); shuitiejunynjkedu@163.com (TS)

**Data Availability Statement:** Meteorological data used for analysis in our manuscript "Retrospective analysis of spatiotemporal variation of scrub typhus in Yunnan Province, 2006-2022" were obtained from the China Meteorological

## Abstract

### Background

Scrub typhus is a life-threatening zoonotic infection. In recent years, the endemic areas of scrub typhus have been continuously expanding, and the incidence rate has been increasing. However, it remains a globally neglected disease. Yunnan Province is a major infected area, and the study of spatiotemporal and seasonal variation scrub typhus in this region is crucial for the prevention and control of the disease.

### Methods/Results

We collected surveillance data on scrub typhus cases in Yunnan Province from 2006 to 2022. Using methods such as spatial trend analysis, Moran's I, and retrospective temporal scan statistics, the spatial and seasonal changes of scrub typhus were analyzed. The study period recorded 71,068 reported cases of scrub typhus in Yunnan Province, with the annual incidence rate sharply increasing ($P<0.001$). Approximately 93.38% of cases are concentrated in June to November ($P = 0.001$). Nearly 98.0% of counties were affected. The center of gravity of incidence migrates in a south and west direction. The incidence of scrub typhus was positively correlated spatially, and the spatial clustering distribution was significant. The most likely spatial cluster of cases (relative risk = 14.09, $P<0.001$) was distributed in Lincang, Dehong, Baoshan, Banna, and Puer. Significant positive correlations between the number of scrub typhus cases and average temperature, precipitation and relative humidity.

### Conclusions

In Yunnan Province, scrub typhus is widely transmitted, with an increasing incidence, and it exhibits distinct seasonal characteristics (from June to November). The center of gravity of incidence has shifted to the south and west, with higher incidence rates observed in border regions. The risk clustering regions encompass all border prefectures. This pattern is

Information Center (http://data.cma.cn/).Access to data requires the applicant to obtain the appropriate meteorological information by registering as a user on the Platform. In addition, data on scrub typhus cases in Yunnan Province from 2006-2022 used for the analysis were obtained from the Yunnan Provincial Disease Prevention and Control Information System. Due to the data involves patients' date of birth, date of onset of illness, home address and other information, we are unable to share it publicly. There are requirements for the use of scrub typhus surveillance data, which requires an application to be submitted to the management of the Yunnan Provincial Center for Disease Control and Prevention (YNCDC) to the administration, describing the organizations and individuals who will use the data, providing the name and organization, and submitting the application to the competent authority (YNCDC Email: yncdc@yncdc. cn). The county-level map of Yunnan Province used in our study was obtained from the Chinese Academy of Sciences, Institute of Geographic Sciences and Natural Resources Research (http:// www.resdc.cn/).This platform allows you to download the required base maps free of charge by registering as a user.

**Funding:** The author(s) received no specific funding for this work.

**Competing interests:** The authors have declared that no competing interests exist.

significantly correlated with climatic factors such as average temperature, precipitation, and relative humidity. The relevant departments should strengthen the monitoring of scrub typhus, formulate prevention and control strategies, and provide health education to local residents.

## Author summary

Scrub typhus is an acute infectious disease of natural epidemic origin caused by *Orientia tsutsugamushi*. Scrub typhus is spread primarily through the bite of a chigger mite larva that carries *Orientia tsutsugamushi*. It's clinical manifestations mainly include non-specific symptoms such as headache, fever, rash, cough, vomiting, muscle pain, and swollen lymph nodes. It can lead to a variety of complications such as pneumonia, encephalitis, and myocarditis, which can be life-threatening in severe cases. Scrub typhus is prevalent globally in Southeast Asia, the Southwest Pacific Islands, Japan, and China, and is expanding. However, it remains a neglected infectious disease. We aim to understand the temporal and seasonal variations of scrub typhus in Yunnan Province, discover the high prevalence areas of scrub typhus and their seasonal patterns, so as to provide a basis for conducting targeted surveillance and health education in Yunnan Province, especially in high-risk areas.

## Introduction

Scrub typhus is a zoonotic disease caused by the *Orientia tsutsugamushi* [1]. Infected larval mite bites are the means by which the pathogen is transmitted [2]. The time it takes for scrub typhus to incubate is around 10–12 days, and its clinical manifestations mainly include non-specific symptoms such as headache, fever, rash, cough, vomiting, muscle pain, and swollen lymph nodes [3]. The most typical feature of scrub typhus is the eschar at the site of the bite, but not all patients develop this symptom (The percentage of patients with eschar ranges from 7% to 97%) [4], making a diagnosis based solely on clinical presentation challenging [5]. Patients generally respond well to antibiotic treatment, but delayed treatment can lead to severe complications and death [6–9]. The median mortality rate for untreated patients is 6.0%, while treated patients have a rate of 1.4% [8, 9].

Globally, there are approximately one million cases of scrub typhus each year, with one billion people at risk of infection [10]. The traditional endemic regions of scrub typhus are referred to as the "tsutsugamushi triangle", covering an area of over 8 million square kilometers, extending from the Far East of Russia to Pakistan, and from Australia to Japan [11]. However, cases of scrub typhus have also been reported in Africa, France, the Middle East, and South America, indicating an expanding endemic area [12]. Despite this, scrub typhus continues to be an overlooked global disease, with many endemic regions lacking research and nationwide surveillance systems. There is a lack of simple, rapid, reliable and cost-effective diagnostic methods and effective protective vaccines [13–15].

In China, scrub typhus now poses a greater threat than ever. Especially in the last two decades, the geographic distribution of scrub typhus in mainland China has expanded dramatically. The incidence rate has risen dramatically from 0.09 per 100,000 population in 2006 to 1.60 per 100,000 population in 2016 [16]. Yunnan Province has the second-highest annual incidence of scrub typhus, with an increasing number of reported cases, and the affected areas expanding [17]. Studies related to the spatial pattern of scrub typhus in Yunnan from 2006–

2013 and 2006–2017 showed that Baoshan, Lincang, and Dehong have the highest number of cases, and study suggests that the number of cases in Yunnan Province may continue to increase in the future [18, 19].

Therefore, the study's goal is to identify scrub typhus hotspots and their seasonal patterns in Yunnan Province, considering spatiotemporal variations and meteorological influences.

## Materials and methods

### Ethics statement

The National Health Commission of China is responsible for collecting scrub typhus case data as part of an ongoing public health monitoring of infectious diseases, hence, it is exempted from the scrutiny of an institutional review board.

### Study area

Yunnan Province is situated between 21˚8' to 29˚15' north latitude and 97˚31' to 106˚11' east longitude. It shares borders with Guizhou and Guangxi to the east, Sichuan to the north, closely adjacent to Tibet in the northwest, Myanmar to the west, and Laos and Vietnam to the south. The total area is 394,100 square kilometers. According to the 2022 Yunnan Provincial Statistical Yearbook, the province has a population of 46.93 million permanent residents and is administratively divided into 16 prefectures, comprising a total of 129 counties (S1 Fig). The base maps of China and Yunnan Province used for the study were obtained from the Institute of Geographic Sciences and Natural Resources, Chinese Academy of Sciences (http://www.resdc.cn/).

### Data collection

In China, scrub typhus cases are reported to the Centers for Disease Control and Prevention (CDC) via the China National Notifiable Infectious Diseases Reporting Information System (CNNDS). All cases of scrub typhus are diagnosed based on the diagnostic criteria guidelines from the CDC in China (S1 Text). Individual-level data of all human scrub typhus cases that occurred in Yunnan province, China, from January 2006 to December 2022, were obtained from the Yunnan Provincial CDC. The data include the patient's current address, date of onset, date of report, reporting institution, date of death, case classification (confirmed, clinically diagnosed, suspected), age, gender, occupation, and other relevant information. The anonymization of all case data used in this study prevents it from revealing the identity of individuals. Patients with suspected cases (S1 Text) were excluded from this study because of uncertainty, and those whose current addresses were outside the Yunnan Province were also excluded. Our analysis includes all cases of clinically human scrub typhus that have been clinically and laboratory-confirmed with current addresses in Yunnan Province (69,799 clinically diagnosed cases and 1269 laboratory-confirmed cases).

Population data for each county in Yunnan Province from 2006 to 2022 were obtained from the Yunnan Provincial Statistical Yearbook (http://stats.yn.gov.cn/).

We collected monthly average temperature (˚C), precipitation (mm), relative humidity (%) for Yunnan province, China. Meteorological surveillance data was obtained from the China Meteorological Information Center (http://data.cma.cn/).

### Data analysis

**Temporal trends analysis.** Microsoft Excel 2016 was used to draw trend charts of case numbers and incidences for different years. The incidence (per 100,000) is determined by

dividing the number of scrub typhus cases in a given year in a given location by the average population of that year. (Since the population at the end of 2022 has not been announced, the average population in 2022 is replaced by the year-end population in 2021). The Mann-Kendall test was employed for detecting trends in time-series data, calculating Kendall rank correlation coefficients, and assessing if there is a positive or negative trend in the scrub typhus incidence and its statistical significance [20]. The Mann-Kendall test was performed with R 4.3.1.

**Seasonal patterns analysis.** Seasonal Temporal analysis was performed with SaTScan v10.1.2. It uses a window that moves in one dimension, time, and with height corresponding to time. The seasonal scan statistic ignores which year the observation was made and only cares about the day and month where all the data is on a connecting loop, such as the year, where December 31 is followed by January 1 [21]. The seasonal distribution was presented in a histogram, plotted in Microsoft Excel 2016.

**Spatial trends analysis.** Calculated the annual incidence (per 100,000) of each county from 2006 to 2022, and used ArcGIS 10.8.2 to create a spatial distribution map.

Using ArcGIS 10.8.2 to correlate each county's incidence data with its latitude and longitude. Then using the Spatial Trend Analysis module. Disease trends were plotted on a three dimensional spatial map. It was to reflect the overall changes in cases in the latitudinal and longitudinal directions of the regional unit [22]. Spatial trend analysis was conducted in the study area. The incidence of each cell was considered as the value of the geometric center of that cell. The analysis used three-dimensional data points $M_i$ ($X_i$, $Y_i$, $Z_i$). In these data points, $X_i$ and $Y_i$ represent the geometric centers of longitude and latitude, respectively, for each area. $Z_i$ denotes the number of cases in that area. Each sampling point $M_i$ was then projected onto the XZ and YZ planes, respectively, to obtain a scatter plot, and a polynomial was fitted through these dispersed points to obtain a curve that was used to model a certain trend of the disease in the area in terms of latitude and longitude [23]. The center of gravity of the annual incidence of scrub typhus was calculated using the center of mean from the Spatial Statistics Tool. Then use the straight line tool to connect each year's center of gravity to form a center of gravity migration trajectory. To analyze the spatial and temporal evolution of scrub typhus morbidity by reflecting the spatial shift of morbidity hotspots through the migration of the center of gravity. The formulas are as follows.

$$\bar{X} = \frac{\sum_{i=1}^{n} S_i X_i}{\sum_{i=1}^{n} S_i}, \; \bar{Y} = \frac{\sum_{i=1}^{n} S_i Y_i}{\sum_{i=1}^{n} S_i}$$

$\bar{X}$ and $\bar{Y}$ denote the coordinates of the center of gravity of the study object (e.g., the number of scrub typhus cases). $S_i$ denotes the value of the subregion of the number of scrub typhus cases, $X_i$ and $Y_i$ denote the latitude and longitude values of the geometric center of the subregion, respectively, and n denotes the number of subregions [24].

**Spatial autocorrelation analysis.** We used ArcGIS 10.8.2 to conduct both global and local spatial autocorrelation analyses. Moran's I was used to detect the existence of global spatial autocorrelation in scrub typhus incidence in Yunnan Province and to measure their correlational strength. A Z-test was used to assess the statistical significance of Moran's I. If Moran's I = 0, it means that there is no spatial autocorrelation and that the incidence of scrub typhus are distributed randomly throughout the study area. Positive spatial autocorrelation is shown by a value of Moran's I > 0, with values nearer 1 suggesting a stronger autocorrelation. Moran's I < 0 denotes a negative spatial autocorrelation, whereas values closer to -1 denote a higher degree of spatial variability [25]. The LISA cluster map displays four types of clusters: high-high (H-H) clusters are places with high incidence surrounded by additional high

incidence areas, low-low (L-L) clusters are places with low incidence surrounded by other low incidence areas, low-high (L-H) clusters are locations with low incidence surrounded by high incidence areas, and high-low (H-L) clusters are places with a high incidence that are surrounded by areas with a low incidence [26].

**Retrospective space-time analysis.**  We used SaTScan v10.1.2 to calculate space-time scan statistics for each county. We used a discrete Poisson model to estimate the relative risk (RR) for each cluster and identified primary and secondary clusters using the log-likelihood ratio (LLR). We assessed cluster significance through 999 Monte Carlo replications. For each model, the input variables included the number of scrub typhus cases, population, and geographical coordinates for each region. We created cylindrical scan windows to perform space-time scanning, where the height of the cylinder represented time and the bottom represented the geographical region. The maximum spatial cluster size was determined to be 10% of the total population, while the maximum temporal cluster size was determined to be 50% of the study time [27, 28]. As each model may identify many statistically significant clusters, we only presented the six most prominent clusters in the figures and tables (the most likely cluster and the five secondary clusters with the highest LLR). We visualized the relative risk of scrub typhus high-incidence areas using ArcGIS 10.8.2.

**Meteorological factors analysis.**  The correlation between meteorological factors and the number of scrub typhus cases was analyzed using Spearman correlation test with IBM SPSS 24.0.

## Results

### Temporal trends

From 2006 to 2022, Yunnan Province reported 71,068 cases of scrub typhus. The annual incidence rose dramatically from 0.65 per 100,000 population in 2006 to 23.94 per 100,000 population in 2022 (Fig 1). This rising trend is evident in all prefectures except Banna (S2 Fig, Detailed parameters are shown in Table 1).

### Seasonal patterns

There is a clear seasonal distribution of scrub typhus in Yunnan Province, with cases beginning to rise rapidly in May, reaching a peak in August, and a rapid decline after November (Fig 2). Approximately 93.38% of cases are concentrated from June to November, with seasonal variations varying slightly from prefecture to prefecture (Table 1).

### Spatial trends

The number of affected counties increased over the years from 38 in 2006 to 127 in 2022. Counties with an annual incidence of more than 18.76/100,000 population are progressively concentrated in the southwestern and southern regions of Yunnan Province, distributed along China's borders with Myanmar, Laos, and Vietnam (Fig 3).

At the same time, the results of the trend surface fitting show that the spatial trend line of the incidence maintains the layout of "high in the south and low in the north, high in the west and low in the east" (Fig 4). The trend surface transition from 2006 to 2007 was predominantly in the north-south direction, while the trend surface transition from 2008 onwards was steeper in the east-west direction than in the north-south direction. The gravity center migration trajectory map also shows that the center of gravity of morbidity migrates in a south and west direction (Fig 5).

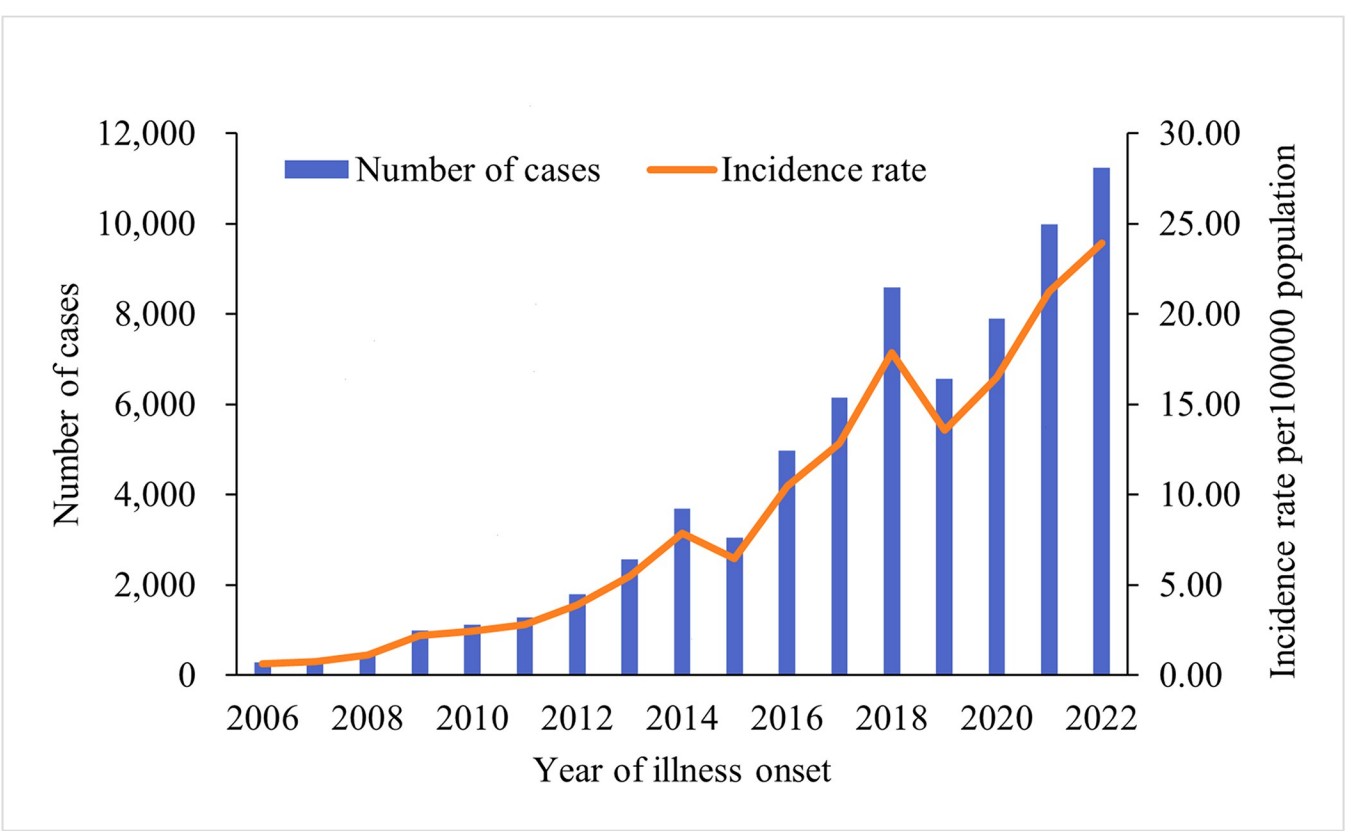

**Fig 1. Reported cases of scrub typhus in Yunnan Province, 2006–2022.** Aggregated number of cases by year (blue bars), annual incidence rate (orange line) per 100,000 population.

**Table 1. Temporal and seasonal trends of scrub typhus in Yunnan Province, 2006–2022.**

| Region | Time trend | P-value | Seasonal Temporal | P-value |
|---|---|---|---|---|
| Diqing | Increase | <0.001 | June to November | 0.001 |
| Nujiang | Increase | <0.001 | July to October | 0.001 |
| Lijiang | Increase | <0.001 | July to October | 0.001 |
| Dali | Increase | 0.015 | July to October | 0.001 |
| Chuxiong | Increase | <0.001 | July to October | 0.001 |
| Kunming | Increase | 0.001 | July to October | 0.001 |
| Qujing | Increase | <0.001 | June to November | 0.001 |
| Zhaotong | Increase | <0.001 | June to October | 0.001 |
| Dehong | Increase | <0.001 | June to November | 0.001 |
| Baoshan | Increase | <0.001 | July to October | 0.001 |
| Lincang | Increase | <0.001 | June to November | 0.001 |
| Puer | Increase | <0.001 | June to November | 0.001 |
| Yuxi | Increase | <0.001 | July to October | 0.001 |
| Honghe | Increase | <0.001 | June to November | 0.001 |
| Wenshan | Increase | <0.001 | June to October | 0.001 |
| Banna | Stable | 0.053 | June to November | 0.001 |

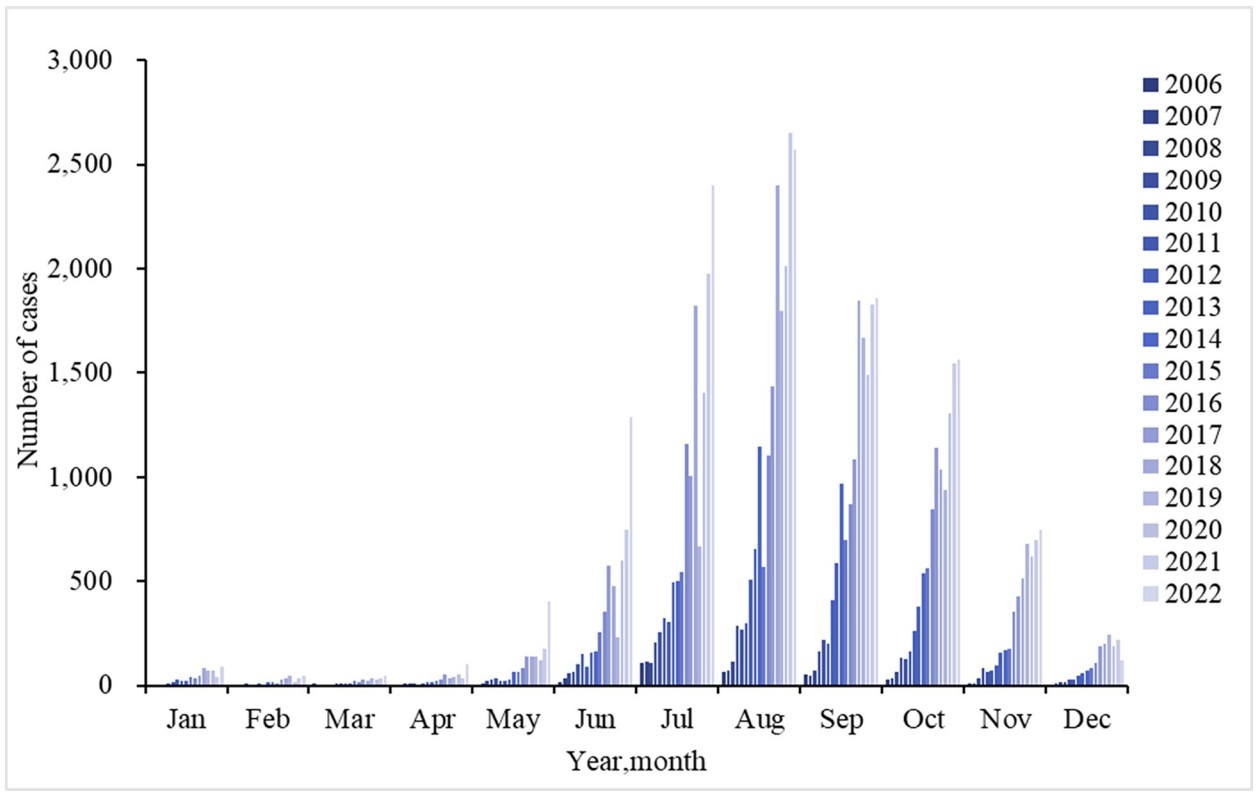

**Fig 2. The seasonal distribution of scrub typhus cases in Yunnan Province.**

### Spatial autocorrelation analysis

The spatial autocorrelation analysis revealed that there was a positive spatial correlation between the incidence of scrub typhus in Yunnan Province from 2007 to 2022, and the spatial clustering distribution was of great significance (Table 2).

The LISA cluster maps showed expanding "high-high" clusters in Yunnan Province's western regions, which were primarily centered there, including Lincang, Dehong, and Baoshan. While "low-low" clusters were growing in the southwestern areas, primarily in these regions, including Dali, Zhaotong, Qujing, Kunming, Yuxi (Huaning, Jiangchuan), Honghe (Luxi, Mile) (S3 Fig).

### Retrospective space-time analysis

Statistical analyses of time-scanning showed that the most likely cluster covered 15 counties distributed in Lincang, Dehong, and Baoshan from 1st July 2014 and 30st November 2022. The secondary clusters comprised 47 counties distributed in Banna, Puer, Honghe, Chuxiong, Wenshan, Yuxi, Kunming, Dali, and Nujiang (Fig 6, Detailed parameters are shown in S1 Table). Notably, the eight border prefectures of Yunnan Province, Nujiang, Baoshan, Dehong, Lincang, Puer, Banna, Honghe, and Wenshan are all included.

### Meteorological factors analysis

The results showed that the peaks in the number of scrub typhus cases were generally consistent with the peaks in average temperature, precipitation, and relative humidity. Correlation analysis showed significant positive correlations between the number of scrub typhus cases

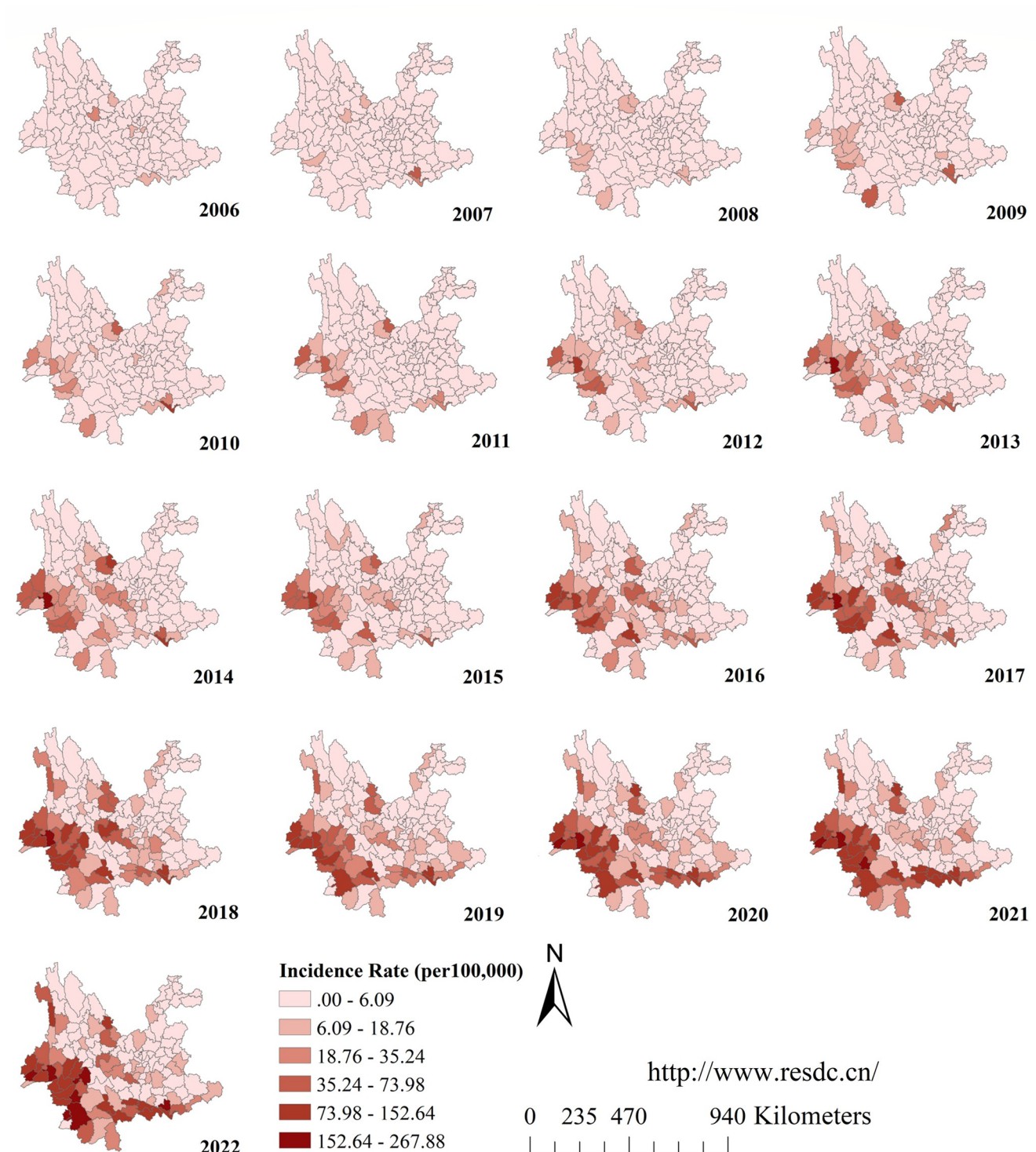

**Fig 3. The geographic distribution of the annual incidence of scrub typhus in Yunnan Province during 2006 to 2022.** (The county-level map of Yunnan Province used in our study was obtained from the Chinese Academy of Sciences, Institute of Geographic Sciences and Natural Resources Research(http://www.resdc.cn/). This platform allows you to download the required base maps free of charge by registering as a user).

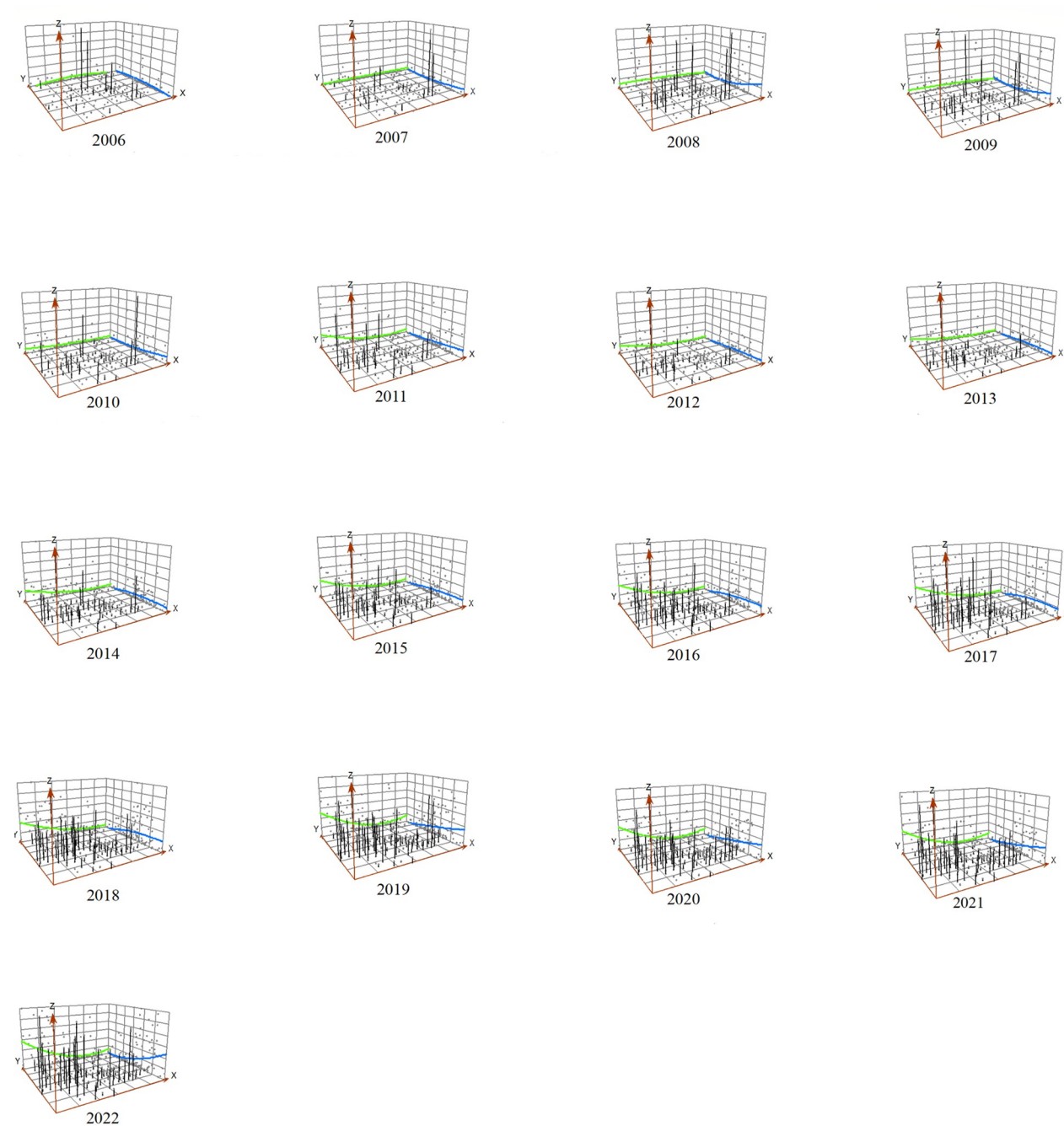

**Fig 4. Spatial trend analysis of the incidence of scrub typhus in Yunnan Province, 2006–2022 (Green line indicates west-east direction, blue line indicates north-south direction).**

and average temperature, precipitation and relative humidity. Spearman correlation coefficients ($r_s$) were 0.522, 0.585, 0.761, respectively (Fig 7).

## Discussion

Based on the analysis of monitoring data for 17 years, we have observed a rapid increase in the incidence of scrub typhus and the number of affected counties in Yunnan Province. The trend

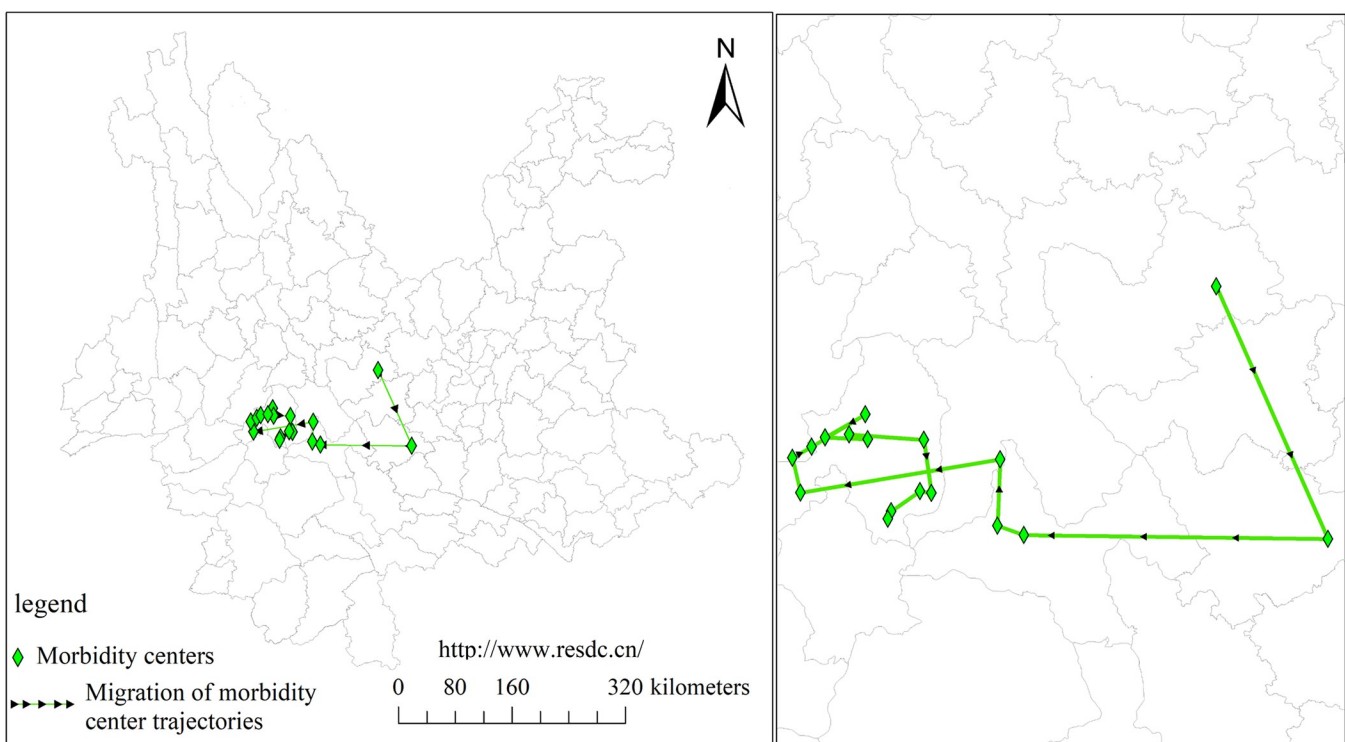

**Fig 5. The gravity center migration trajectory of scrub typhus in Yunnan Province from 2006 to 2022.** (The county-level map of Yunnan Province used in our study was obtained from the Chinese Academy of Sciences, Institute of Geographic Sciences and Natural Resources Research(http://www.resdc.cn/). This platform allows you to download the required base maps free of charge by registering as a user).

**Table 2. Global spatial autocorrelation analysis of scrub typhus incidence in Yunnan Province, China, 2006–2022.**

| Year | Moran's $I$ | Z-score | P-value | Cluster |
|------|-----------|---------|---------|---------|
| 2006 | -0.019 | -0.277 | 0.782 | NO |
| 2007 | 0.197 | 4.440 | <0.001 | Yes |
| 2008 | 0.243 | 4.594 | <0.001 | Yes |
| 2009 | 0.164 | 3.246 | 0.001 | Yes |
| 2010 | 0.159 | 3.314 | <0.001 | Yes |
| 2011 | 0.259 | 4.944 | <0.001 | Yes |
| 2012 | 0.234 | 5.408 | <0.001 | Yes |
| 2013 | 0.247 | 5.714 | <0.001 | Yes |
| 2014 | 0.290 | 5.606 | <0.001 | Yes |
| 2015 | 0.441 | 8.110 | <0.001 | Yes |
| 2016 | 0.453 | 8.223 | <0.001 | Yes |
| 2017 | 0.456 | 8.169 | <0.001 | Yes |
| 2018 | 0.462 | 8.213 | <0.001 | Yes |
| 2019 | 0.496 | 8.808 | <0.001 | Yes |
| 2020 | 0.457 | 8.161 | <0.001 | Yes |
| 2021 | 0.490 | 8.791 | <0.001 | Yes |
| 2022 | 0.492 | 8.705 | <0.001 | Yes |

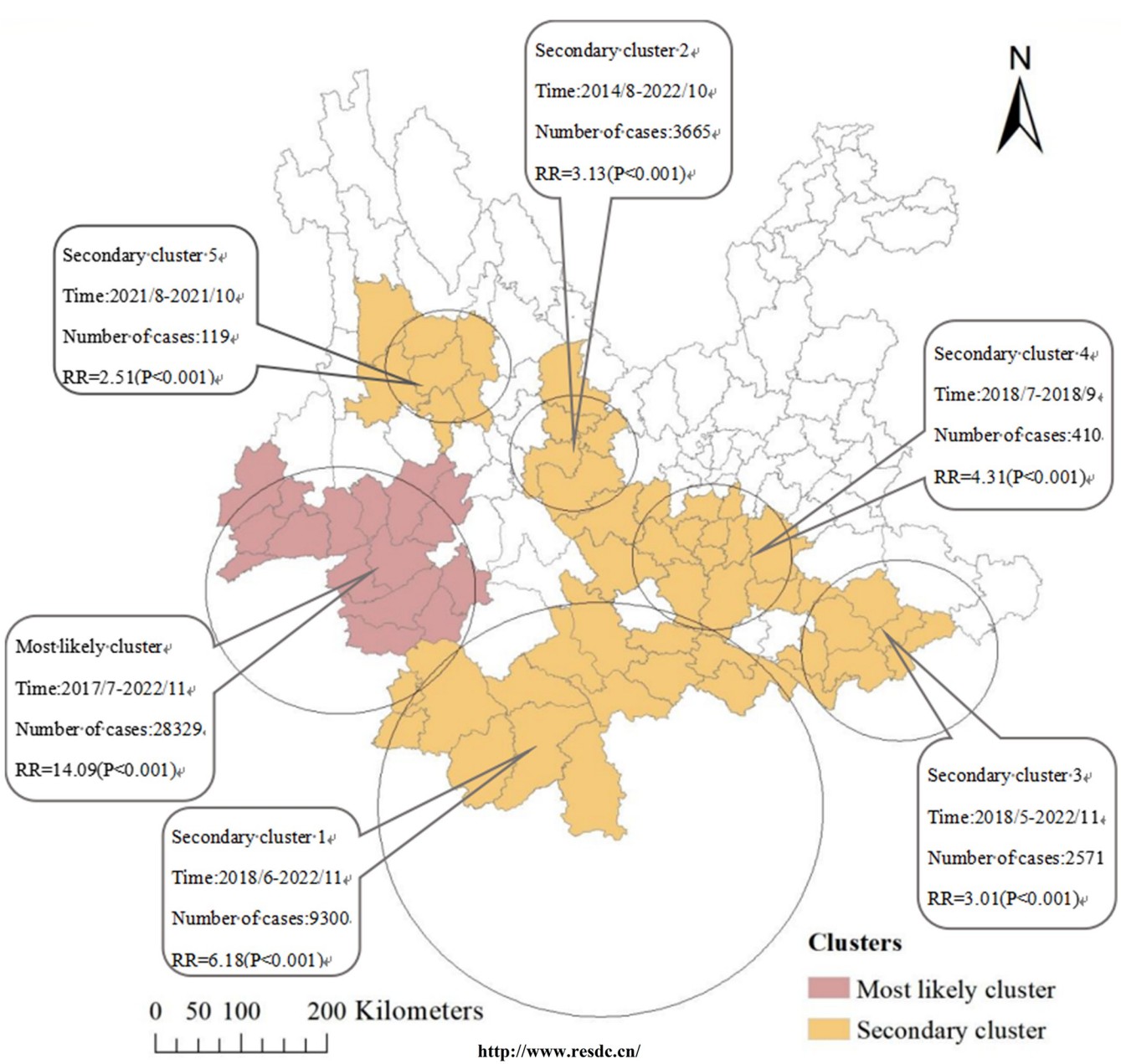

**Fig 6. Spatiotemporal clusters of scrub typhus in Yunnan Province, 2006–2022.** Light redindicates the most likely high-risk spatiotemporal clustering area, orange indicates the secondary-risk clustering area. (The county-level map of Yunnan Province used in our study was obtained from the Chinese Academy of Sciences, Institute of Geographic Sciences and Natural Resources Research (http://www.resdc.cn/). This platform allows you to download the required base maps free of charge by registering as a user).

suggests that the incidence of scrub typhus might continue to rise in the future. From 2006 to 2022, the incidence of scrub typhus in Yunnan Province increased from 0.65 per 100,000 population to 23.94 per 100,000 population, far higher than the national average. The incidence rate of scrub typhus in mainland China increased from 0.09 to 1.60 per 100,000 population between 2006 and 2016 [15].

Scrub typhus has been monitored through the CNNDS, a web-based, real-time, direct reporting system for infectious diseases since 2006. The use of passive surveillance systems and

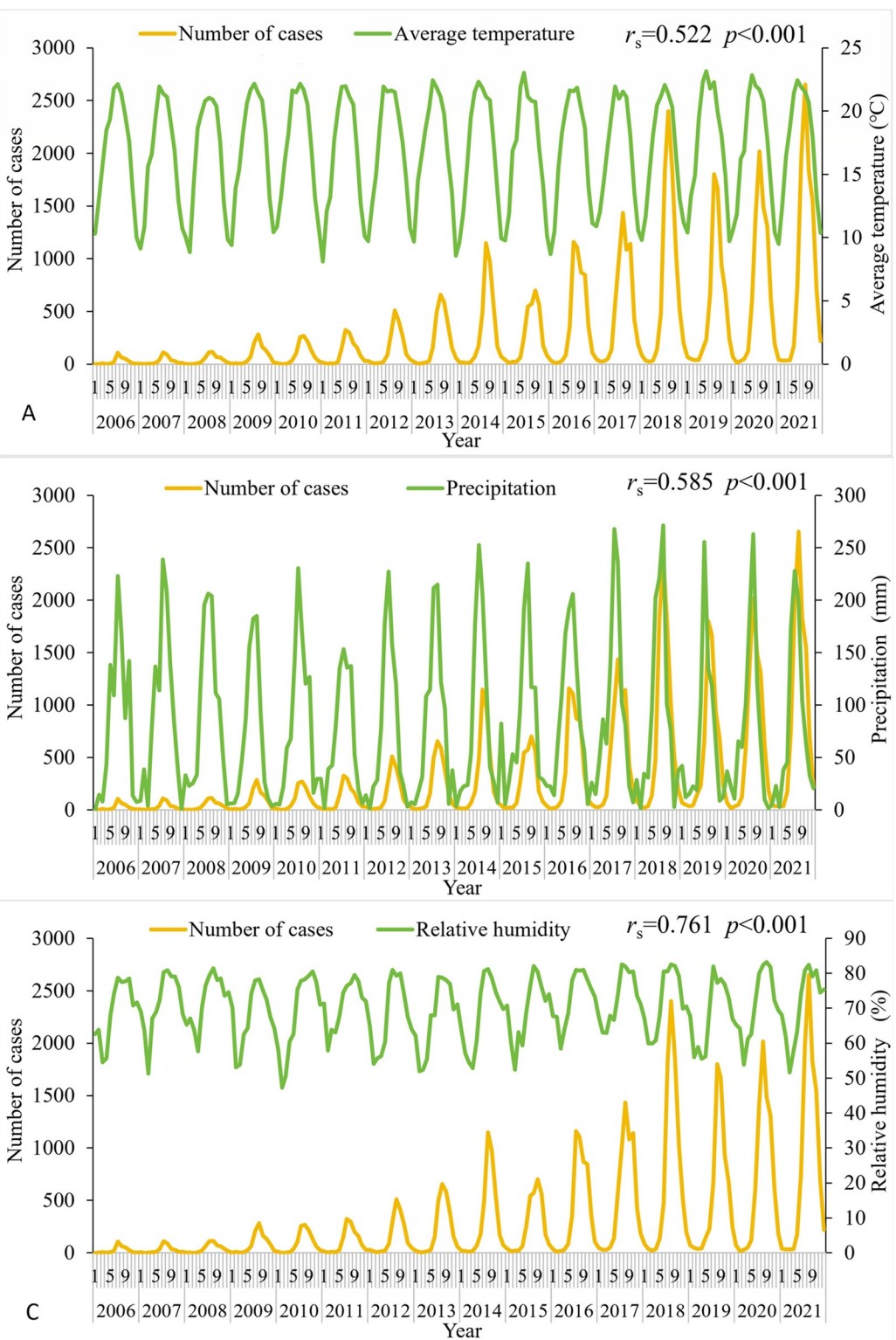

**Fig 7. The relationship between meteorological factors and the number of scrub typhus cases in Yunnan Province, 2006–2022.**

the increased use of diagnostic tools may lead to an increase in scrub typhus reporting rates. This could be one reason for the increase in scrub typhus incidence in Yunnan Province [15].

Chigger mites, typically parasitic in small rodents, are the exclusive vector of scrub typhus [29, 30]. Yunnan Province has a significantly higher population of chigger mites compared to other Chinese provinces and even surpasses the number of chigger mite species in some global regions and countries. Additionally, the diversity of small mammal host species in Yunnan is notably high [31]. Disease vectors and hosts may move due to human behavior and changes in the social environment, potentially spreading pathogens to distant areas and causing disease transmission [32, 33]. In recent years, accelerated urbanization, changes in the natural environment, the development of tourism and climate change in Yunnan Province have created favorable conditions for the spread of rodent hosts and mites, increasing the likelihood of human exposure to these hosts and mites [34]. This may be one of the reasons for the high incidence of scrub typhus in Yunnan Province.

Scrub typhus in Yunnan Province exhibits distinct seasonal characteristics with a peak from June to November, belonging to the summer-autumn type. This is different from the autumn-winter type in northern China and the double-peak pattern in southeastern regions. Other countries with scrub typhus outbreaks have reported varying seasonal patterns as well, such as spring-summer peaks in northern Japan and autumn-winter peaks in the south. In Korea, the epidemic period occurs from October to November, and 97% of the cases in Chile occur in the summer months from December to March [35–37]. The geographic variation in the distribution of chigger mites, which carry scrub typhus, may contribute to the observed differences in seasonal patterns. While *Leptotrombidium scutellare* primarily manifests in winter and spring, with the peak coming from January to February, the dominant species in Yunnan, *Leptotrombidium deliense* peaks during summer and autumn, with the biggest peak occurring in July [38]. Scrub typhus's seasonal distribution may also change as a result of intricate interactions between vectors, hosts, and people as well as climatic variables. Our analysis shows that scrub typhus cases in Yunnan Province are associated with higher temperatures, relative humidity, and precipitation. Yunnan has abundant rainfall, high humidity and temperature in summer and fall, which is suitable for the growth of chiggers. This may also contribute to the seasonality of scrub typhus.

Spatial trend analysis showed that the southern and southwestern regions of Yunnan Province (covering border areas) had a higher incidence of scrub typhus, and had become the main areas where the center of gravity of scrub typhus had shifted. Similarly, spatial autocorrelation analyses showed that the scrub typhus "high-high" clusters are primarily located in the southwestern regions of Yunnan Province, including Lincang, Dehong, and Baoshan. This may be related to the local climate. A study has revealed positive correlations between temperature, relative humidity, precipitation, and scrub typhus incidence [39]. This is consistent with our results. Our results showed that the number of scrub typhus cases was significantly and positively correlated with average temperature, precipitation and relative humidity. Another study shows that temperature, humidity, and rainfall have a nonlinear impact and a lag effect on the prevalence of scrub typhus in southwest Yunnan [40]. High temperatures, high humidity, and heavy rainfall increase the risk of scrub typhus [40]. Southern and southwestern Yunnan is in the low and hot valley with low latitude and warm climate, while parts of the region south of the Tropic of Cancer are in the tropical range, where temperatures are high and rainfall is abundant. The environment in these areas provides favorable conditions for the growth of chiggers. Research on a region including Yunnan showed that in low altitude and low latitude areas, the *Rattus tanezumi* is the main parasitic host of chigger mites [41]. Warm weather and high humidity are conducive to chigger infestation on Rattus tanezumi. Climate change is a long and slow process, and the magnitude of change over a short period is not obvious, but the

enormous impact of climate change is an indisputable fact. Many studies have shown that there is a clear trend of climate warming in Yunnan Province, with greater warming in western and southwestern Yunnan, and a general upward trend in annual precipitation intensity [42–44]. This implies that alterations in meteorological factors could significantly impact the long-term trend of scrub typhus epidemics in Yunnan Province.

In addition, changes in natural environmental factors and socio-economic factors can have an impact on the prevalence of scrub typhus. Higher scrub typhus incidence is associated with higher Normalized Difference Vegetation Index (NDVI) [45]. Changes in land use land cover (LULC) and urbanization may lead to an upsurge in scrub typhus cases [46]. In recent years, policies and measures in southern and southwestern Yunnan Province, such as heavy investment in transportation infrastructure such as highways, railroads, and airports, and the development of specialty industries such as tea, coffee, and medicinal herbs, may have led to changes in land cover and increased human contact with chiggers. This has led to an increase in the incidence of scrub typhus in these areas. It's worth noting that there are "low-high" clusters every year, indicating that monitoring and prevention efforts should not only focus on "high-high" clusters but also consider the "low-high" clusters in the surrounding areas to prevent them from becoming a new "high".

Similarly, studies have shown that scrub typhus in northern Vietnam is prevalent in border areas and may be related to host-vector status [47]. In the northeastern Indian state of Mizoram, which borders Myanmar and Bangladesh, scrub typhus has also emerged as a major local health problem, linked to the state's rich biodiversity and high forest cover [48]. Chiggers may be spread in the border areas of Laos and Thailand through the trading activities, commuting or traveling of villagers in the border areas. This may be due to human movement and the movement of mites through animals [49]. Given that Yunnan's neighboring countries are also endemic for scrub typhus, cross-border travelers and workers should pay attention to prevention and control to avoid contracting scrub typhus during their journeys.

The results of retrospective spatio-temporal analyses showed that clustering areas includes 62 counties in Lincang, Dehong, Baoshan, Banna, Puer, Honghe, Chuxiong, Wenshan, Yuxi, Kunming, Dali, and Nujiang. The results mentioned above are the same as the findings of local spatial autocorrelation analysis, but not entirely identical. Spatial-temporal scan statistics have significant advantages, such as scale selection, scale transformation, spatiotemporal integration, and quantitative evaluation of potential clusters and their sizes. These methods help identify the most significant high-risk clusters, providing valuable references for formulating and improving prevention and control strategies [50].

Our study has some limitations because the data is derived from the surveillance data of the Yunnan Provincial Center for Disease Control and Prevention, which does not reveal the laboratory methods used in case diagnosis. The current use of different laboratory methods exhibits inconsistent specificity and sensitivity, particularly the specificity and sensitivity of the Weil Felix test are not high enough [5]. This could result in some false positive and false negative rates [15]. In addition, scrub typhus has been included in CNNDS since 2006. But scrub typhus is classified as "other" infectious diseases and is not included in the list of legal infectious diseases in mainland China, leading to underreporting issues. These may lead to discrepancies between our data and the real situation. However, the information used in this study is the most thorough and trustworthy information available on scrub typhus in Yunnan Province. Moreover, given this study's goal was to compare incidence trends and geographical distribution, it can be reasonably assumed that the rate of false positives and underreporting is similar across different regions and populations. Thus, the research results are unlikely to be significantly affected by reporting deficiencies.

Our study is the first to analyze the migration of the center of gravity of the incidence of scrub typhus in Yunnan Province, and is the most up-to-date and comprehensive spatio-temporal analysis of scrub typhus in Yunnan Province. Our study found that scrub typhus is widespread in Yunnan Province, with a rapidly increasing incidence, a shift in the center of gravity of the disease to the south and west, a high incidence in prefectures distributed along the borderline, and the existence of areas of high clusters and possible risk clusters. These findings underscore that scrub typhus has grown to be a serious public health concern in epidemic-prone areas. To cope with the new situation, the management should strengthen its monitoring, prevention and control efforts in the whole Yunnan Province, especially in the southern and southwestern regions. Health education should be provided to people living in areas where the disease is endemic, especially for the elderly and children, raising their awareness of diseases enables them to take preventive measures during outdoor activities. Elderly people and children, who have a high incidence of scrub typhus in Yunnan Province, are also the focus of health education because of their relatively low knowledge base, lack of health awareness, and more outdoor farm work [51]. During the epidemic season of scrub typhus, the authorities concerned should formulate a strategy for the control of scrub typhus and travelers should take precautionary measures before entering areas where scrub typhus is endemic. Early diagnosis and treatment of this disease can greatly reduce mortality; therefore, healthcare workers should be sensitized to this disease and consider the possibility of scrub typhus when unexplained fever occurs in highly endemic areas. Subsequent research should further explore the reasons and spatiotemporal distribution changes that affect the transmission of scrub typhus from the perspectives of host, environment, and socio-economic factors.

## Conclusions

In Yunnan Province, scrub typhus is widely transmitted, with an increasing incidence, and it exhibits distinct seasonal characteristics (from June to November). The center of gravity of incidence shifted to the south and west, and the border regions show higher incidence. The risk clustering regions cover all border prefectures. Significant positive correlations between the number of scrub typhus cases and average temperature, precipitation and relative humidity. The relevant departments should strengthen the monitoring of scrub typhus, formulate prevention and control strategies, and provide health education to local residents.

## Supporting information

**S1 Fig. Map of the study area.** (The county-level map of Yunnan Province used in our study was obtained from the Chinese Academy of Sciences, Institute of Geographic Sciences and Natural Resources Research(http://www.resdc.cn/). This platform allows you to download the required base maps free of charge by registering as a user).
(TIF)

**S2 Fig. Incidence rate of scrub typhus in 16 prefectures of Yunnan Province.**
(TIF)

**S3 Fig. The LISA cluster map.** (The county-level map of Yunnan Province used in our study was obtained from the Chinese Academy of Sciences, Institute of Geographic Sciences and Natural Resources Research(http://www.resdc.cn/). This platform allows you to download the required base maps free of charge by registering as a user).
(TIF)

**S1 Text. Diagnostic criteria for scrub typhus.**
(DOCX)

**S1 Table. Space-time scan statistical analysis of scrub typhus in Yunnan Province, 2006–2022.**
(DOCX)

**S1 Data. Data of scrub typhus cases in Yunnan Province, 2006–2022.**
(DOCX)

## Acknowledgments

We gratefully thank the staff members of the Center for Disease Control and Prevention of Yunnan for their assistance in the data collection, and validation.

## Author Contributions

**Conceptualization:** Zhuo Li, Shuzhen Deng, Qing Zhen, Tiejun Shui.

**Data curation:** Zhuo Li, Shuzhen Deng, Tian Ma, Jiaxin Hao, Hao Wang, Xin Han, Menghan Lu, Shanjun Huang, Dongsheng Huang, Shuyuan Yang.

**Formal analysis:** Hao Wang.

**Methodology:** Zhuo Li, Shuzhen Deng, Menghan Lu.

**Software:** Tian Ma, Jiaxin Hao.

**Supervision:** Dongsheng Huang, Shuyuan Yang, Qing Zhen, Tiejun Shui.

**Visualization:** Xin Han, Shanjun Huang.

**Writing – original draft:** Zhuo Li, Shuzhen Deng.

**Writing – review & editing:** Zhuo Li, Shuzhen Deng, Tian Ma, Jiaxin Hao, Hao Wang, Xin Han, Menghan Lu, Shanjun Huang, Dongsheng Huang, Shuyuan Yang, Qing Zhen, Tiejun Shui.

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
