## [Decision Letter · Decision Letter 0]

2 May 2024

Dear Dr. Zhen,

Thank you very much for submitting your manuscript "Retrospective analysis of spatiotemporal variation of scrub typhus in Yunnan Province, 2006-2022" for consideration at PLOS Neglected Tropical Diseases. As with all papers reviewed by the journal, your manuscript was reviewed by members of the editorial board and by several independent reviewers. The reviewers appreciated the attention to an important topic. Based on the reviews, we are likely to accept this manuscript for publication, providing that you modify the manuscript according to the review recommendations. 

Dear Authors

Thank you for your submission to PLoS NTD. In order for us to accept your manuscrip for publication, please see the reviewers' comments attached. Please address all the comments put forth by the reviewers, and make the necessary changes/edits where appropriate.

Best regards

Sincerely,

Yazid Abdad

Guest Editor

Nigel Beebe

Section Editor

Dear Authors

Thank you for your submission to PLoS NTD. In order for us to accept your manuscrip for publication, please see the reviewers' comments attached. Please address all the comments put forth by the reviewers, and make the necessary changes/edits where appropriate.

Best regards

Reviewer's Responses to Questions

**Key Review Criteria Required for Acceptance?**

**Methods**

-Are the objectives of the study clearly articulated with a clear testable hypothesis stated?

-Is the study design appropriate to address the stated objectives?

-Is the population clearly described and appropriate for the hypothesis being tested?

-Is the sample size sufficient to ensure adequate power to address the hypothesis being tested?

-Were correct statistical analysis used to support conclusions?

-Are there concerns about ethical or regulatory requirements being met?

Reviewer #1: The study describes a spatio temporal analysis using surveillance data from the Yunnan province in China. The study design, the population described, sample size and statistical analysis seem adequate to consider the manuscript for publication.

Reviewer #2: Although often used for diagnosis, the Weil-Felix test has poor specificity. Please include a statement of this as a limitation of the data. It would also be helpful to include the proportion of results from this method and how this taken into account in defining cases.

Reviewer #3: (No Response)

**Results**

-Does the analysis presented match the analysis plan?

-Are the results clearly and completely presented?

-Are the figures (Tables, Images) of sufficient quality for clarity?

Reviewer #1: The results presented are as per analysis plan and the presentation of images are of sufficient clarity.

Reviewer #2: Results are presented well.

Reviewer #3: (No Response)

**Conclusions**

-Are the conclusions supported by the data presented?

-Are the limitations of analysis clearly described?

-Do the authors discuss how these data can be helpful to advance our understanding of the topic under study?

-Is public health relevance addressed?

Reviewer #1: The limitations described and conclusions are adequate and the public health importance of the study has been adequately addressed.

Reviewer #2: Please include the proportion of the positive results received by Weil Felix test, and address the limitation of the use of this test due to its low specificity and how this was accounted for in the analysis.

Reviewer #3: (No Response)

**Editorial and Data Presentation Modifications?**

Reviewer #1: There are a few spelling errors such as the word 'shrub typhus' used instead of 'scrub typhus'.

Reviewer #2: none

Reviewer #3: (No Response)

**Summary and General Comments**

Reviewer #1: Overall, the study uses secondary data to analyse the spatio-temporal distribution of scrub typhus in the Yunnan province in China over a period of 16 years. The study is well conducted and presented.

Reviewer #2: The authors do a good job of describing the incidence of scrub typhus in the Yunnan Province, from 2006-2022 with relevant discussion of the analysis. The lab methods in use are standard testing methods of varying specificities and the data could be strengthened by including these as a limitation (specifically regarding the Weil Felix test) or acknowledging how this was taken int account for case definition and analysis.

Reviewer #3: This paper presents a relatively standard analysis of scrub typhus case data from Yunnan Province of China. As is, I cannot tell what we've learned from this analysis. Some simple statistical approaches are applied to case data across administrative units and over time. There is an argument for some novelty in the measurement of the "migration of the center of gravity of scrub typhus). However, it isn't clear why this is an important thing to do. It also appears to be a geostatistical approach that is meant for point data that is here being applied to geometric centroids of administrative units. While this latter critique may be minor, it should be addressed, and I stress that it isn't clear why it is important to look at the movement of the 'center of gravity'. Case data are already being reported throughout this region - it seems best to keep doing that rather than try to focus more case detection based on this statistic that might be spuriously moving because of the shapes of the administrative units. 

Also, it is curious that scrub typhus is not a reported disease in this setting, but that the authors come to a conclusion that changes in reporting cannot be driving the patterns that are evident in space and time. 

Finally, this paper would seriously benefit from further analyses and contextualization. It seems that cases are occurring more frequently in the 'summer'. Does this correspond to the rainy season? How about an analysis that looks at rainfall patterns, or surface water (NDWI, etc.), and reported case incidence? How about looking at other environmental factors? Descriptive epidemiology is important, especially for under-studied diseases, but this paper currently reads as though it was a report being written for local health authorities.

PLOS authors have the option to publish the peer review history of their article (what does this mean?). If published, this will include your full peer review and any attached files.

Reviewer #1: Yes: Winsley Rose

Reviewer #2: No

Reviewer #3: No

Figure Files:

Data Requirements:

Reproducibility:

References

---

## [Decision Letter · Decision Letter 1]

30 Jul 2024

Dear Dr. Zhen,

Thank you very much for submitting your manuscript "Retrospective analysis of spatiotemporal variation of scrub typhus in Yunnan Province, 2006-2022" for consideration at PLOS Neglected Tropical Diseases. As with all papers reviewed by the journal, your manuscript was reviewed by members of the editorial board and by several independent reviewers. The reviewers appreciated the attention to an important topic. Based on the reviews, we are likely to accept this manuscript for publication, providing that you modify the manuscript according to the review recommendations. 

Dear Authors,

Thank you for resubmitting the manuscript with the changes made after the first review. We deemed it necessary to review it again to ensure that the quality fo the submitted manuscript is in line with our requirements. There are some new recommendations from the reviewers that I hope you can address. Please make the necessary changes where appropriate and address them accordingly.

Sincerely,

Yazid Abdad

Guest Editor

Nigel Beebe

Section Editor

Dear Authors,

Thank you for resubmitting the manuscript with the changes made after the first review. We deemed it necessary to review it again to ensure that the quality fo the submitted manuscript is in line with our requirements. There are some new recommendations from the reviewers that I hope you can address. Please make the necessary changes where appropriate and address them accordingly.

Reviewer's Responses to Questions

**Key Review Criteria Required for Acceptance?**

**Methods**

-Are the objectives of the study clearly articulated with a clear testable hypothesis stated?

-Is the study design appropriate to address the stated objectives?

-Is the population clearly described and appropriate for the hypothesis being tested?

-Is the sample size sufficient to ensure adequate power to address the hypothesis being tested?

-Were correct statistical analysis used to support conclusions?

-Are there concerns about ethical or regulatory requirements being met?

Reviewer #2: The authors have provided sufficient adjustments to the manuscript from previous reviewer comments

Reviewer #4: (No Response)

**Results**

-Does the analysis presented match the analysis plan?

-Are the results clearly and completely presented?

-Are the figures (Tables, Images) of sufficient quality for clarity?

Reviewer #2: The authors have provided sufficient adjustments to the manuscript from previous reviewer comments

Reviewer #4: (No Response)

**Conclusions**

-Are the conclusions supported by the data presented?

-Are the limitations of analysis clearly described?

-Do the authors discuss how these data can be helpful to advance our understanding of the topic under study?

-Is public health relevance addressed?

Reviewer #2: In the discussion, authors describe “healthcare workers should be sensitized to this disease and consider the possibility of scrub typhus when unexplained fever occurs in highly endemic areas”. Please consider and address the need for education and awareness efforts at the community level. Are there public health awareness efforts? If so, what are they and please state the possible effect of these efforts.

Reviewer #4: (No Response)

**Editorial and Data Presentation Modifications?**

Reviewer #2: Line 106. Describe what is meant by “suspicious cases”

Line 106-108. Were travel related cases considered/counted as where they may have become infected or only where the patient held residence?

Line 109-110. Please state the proportion of cases that were diagnosed clinically, and laboratory confirmed.

Reviewer #4: (No Response)

**Summary and General Comments**

Reviewer #2: The authors have done a good job responding to reviewer's comments. Please consider the following general comments.

Line 28. 29. Please use Italics with O. tsutsugamushi

Line 44. Replace “triggered” with “caused by” 

Lines 48-50. Describe/state the percentage of patients have eschars?

Line 62-63. Describe what is considered “rapid and timely diagnostic methods” and how is it different from existing methods like PCR?

Line 65-67. Check redundancy in sentence.

Line 72. Why do you expect to see an increase in Yunnan Providence as indicated?

Reviewer #4: • The abstract lacks a conclusion part and needs to include one during the submission process.

• Line 95: "Data collection" should be adjusted as a sub-title.

• Line 109: "Province" should be written as "province."

• Line 113: The link provided does not work.

• Line 115: Change "from" to "for."

• Figure S1: Suggest rewording the legend name “states.”

• Line 122: Suggest making it clearer how the 2022 population was calculated.

• Line 138: If there is no specific method or reason for these classes, please remove this sentence as readers can understand it through the map.

• Line 141: Reword to "Then, using the Spatial Trend Analysis module, disease trends were plotted on a three-dimensional spatial map to reflect the overall changes in cases in both the latitudinal and longitudinal directions of the regional unit."

• Figure 1: Suggest capitalizing the legend name “Incidence rate.”

• Figure 5: Legend name not capitalized.

• Results, Meteorological factors analysis: It seems the authors only conducted a Spearman correlation test between average temperature, precipitation, relative humidity, and scrub typhus cases. Suggest using more advanced methods to explore their associations on a finer spatial scale, like the county level. Additionally, consider expanding the analysis to include geographic and socio-economic factors. The regional heterogeneity could be driven and explained by multiple factors, which would be interesting to see.

• Since Yunnan province shares borders with Myanmar, Laos, and Vietnam, it would be valuable to include results from imported cases.

• Discussion, lines 259 to 268: Although the authors mentioned that the overall incidence of reported infectious diseases in China remained relatively stable, this cannot lead to the conclusion that changes in reporting do not explain the variation of scrub typhus in Yunnan. The overall stable incidence of infectious diseases reported nationwide does not necessarily mean that the reporting rate of scrub typhus in a specific region (Yunnan) is also stable. Suggest reconsidering this paragraph.

• Line 306: Lack of reference.

• Line 305: "study" should be lowercase, not "Study."

• Discussion, lines 299 to 323: These lines all refer to other studies without describing this study’s association analysis. Suggest adding more analysis on how the county-level case numbers relate to local meteorological and other factors to enrich the results and support the discussion.

• Discussion, lines 358-361: Need reference.

• Lines 361-363: To my knowledge, in 2006, scrub typhus was added as a reporting disease in the National Notifiable Infectious Disease Reporting Information System at the Chinese Centers for Disease Control and Prevention (China CDC). Please clarify this.

• Line 369: "In general, the study's findings can represent the real situation of this disease in Yunnan Province." Suggest deleting this. It is not proper to say this after a whole paragraph of limitations.

• Since the study period covered the COVID-19 period, suggest adding a statement about whether there was any impact of COVID-19.

• Why is the title of S6 "date"? Should it be "data"?

PLOS authors have the option to publish the peer review history of their article (what does this mean?). If published, this will include your full peer review and any attached files.

Reviewer #2: No

Reviewer #4: No

Figure Files:

Data Requirements:

Reproducibility:

References

---

## [Decision Letter · Decision Letter 2]

25 Oct 2024

Dear Dr. Zhen,

We are pleased to inform you that your manuscript 'Retrospective analysis of spatiotemporal variation of scrub typhus in Yunnan Province, 2006-2022' has been provisionally accepted for publication in PLOS Neglected Tropical Diseases.

Best regards,

Yazid Abdad

Guest Editor

Nigel Beebe

Section Editor

Dear Authors,

Please address the suggestions and recommendations made by the reviewers. Please address each point made where applicable and the changes suggested.

Best regards

Reviewer's Responses to Questions

**Key Review Criteria Required for Acceptance?**

**Methods**

-Are the objectives of the study clearly articulated with a clear testable hypothesis stated?

-Is the study design appropriate to address the stated objectives?

-Is the population clearly described and appropriate for the hypothesis being tested?

-Is the sample size sufficient to ensure adequate power to address the hypothesis being tested?

-Were correct statistical analysis used to support conclusions?

-Are there concerns about ethical or regulatory requirements being met?

Reviewer #4: (No Response)

Reviewer #5: The methodology part was very good, straightforward and easy to understand.

**Results**

-Does the analysis presented match the analysis plan?

-Are the results clearly and completely presented?

-Are the figures (Tables, Images) of sufficient quality for clarity?

Reviewer #4: (No Response)

Reviewer #5: Results presented very well with tables, figures, and maps to understand.

**Conclusions**

-Are the conclusions supported by the data presented?

-Are the limitations of analysis clearly described?

-Do the authors discuss how these data can be helpful to advance our understanding of the topic under study?

-Is public health relevance addressed?

Reviewer #4: (No Response)

Reviewer #5: The conclusion was acceptable; however, it was separated from the last paragraph of the discussion. Please clarify the conclusion.

**Editorial and Data Presentation Modifications?**

Reviewer #4: (No Response)

Reviewer #5: (No Response)

**Summary and General Comments**

Reviewer #4: (No Response)

Reviewer #5: While the objectives, methodology, and results are clearly presented, there are some issues with the write-up. Specific comments have been provided in the attachment for your review. Suggest to send for proofread.

PLOS authors have the option to publish the peer review history of their article (what does this mean?). If published, this will include your full peer review and any attached files.

Reviewer #4: No

Reviewer #5: No

---

## [Editor Report · Acceptance letter]

12 Nov 2024

Dear Dr. Zhen,

We are delighted to inform you that your manuscript, "Retrospective analysis of spatiotemporal variation of scrub typhus in Yunnan Province, 2006-2022," has been formally accepted for publication in PLOS Neglected Tropical Diseases.

Best regards,

Shaden Kamhawi

co-Editor-in-Chief

Paul Brindley

co-Editor-in-Chief
